# Biological Predictors of Treatment Response in Adult Attention Deficit Hyperactivity Disorder (ADHD): A Systematic Review

**DOI:** 10.3390/jpm12101742

**Published:** 2022-10-20

**Authors:** Enrico Capuzzi, Alice Caldiroli, Anna Maria Auxilia, Riccardo Borgonovo, Martina Capellazzi, Massimo Clerici, Massimiliano Buoli

**Affiliations:** 1Psychiatric Department, Azienda Socio Sanitaria Territoriale Monza, 20900 Monza, Italy; 2Department of Medicine and Surgery, University of Milano Bicocca, 20900 Monza, Italy; 3Department of Pathophysiology and Transplantation, University of Milan, 20122 Milan, Italy; 4Department of Neurosciences and Mental Health, Fondazione IRCCS Ca’ Granda Ospedale Maggiore Policlinico, 20122 Milan, Italy

**Keywords:** attention-deficit/hyperactivity disorder (ADHD), adult, markers, treatment response, outcome

## Abstract

Background: Attention-deficit/hyperactivity disorder (ADHD) is a highly prevalent condition with onset in childhood and in many cases persisting into adulthood. Even though an increasing number of studies have investigated the efficacy of pharmacotherapy in the management of adult ADHD, few authors have tried to identify the biological predictors of treatment response. Objectives: To summarize the available data about the biological markers of treatment response in adults affected by ADHD. Methods: A search on the main biomedical and psychological archives (PubMed, Embase, Scopus, and PsycINFO) was performed. Manuscripts in English, published up to May 2022 and having the biological predictors of treatment response in adults with ADHD as their main topic, were included. Results: A total of 3855 articles was screened. Twenty-two articles were finally included. Most of the manuscripts studied neuroimaging and electrophysiological factors as potential predictors of treatment response in adult ADHD patients. No reliable markers were identified until now. Promising findings on this topic regard genetic polymorphisms in snap receptor (SNARE) proteins and default mode network-striatum connectivity. Conclusions: Even though some biological markers seem promising for the prediction of treatment response in adults affected by ADHD, further studies are needed to confirm the available data in the context of precision medicine.

## 1. Introduction

Attention-deficit hyperactivity disorder (ADHD) is a neurodevelopmental disorder with onset in childhood, albeit up to 65% of affected children may continue to manifest symptoms in adulthood [1]. Indeed, most cases demonstrated fluctuating symptoms between childhood and young adulthood. A very recent longitudinal study by Sibley (2022) reported that more than half of children (*n* = 558) experience a recurrence of ADHD after an initial period of remission. Moreover, only 10% of children presented a sustained remission or demonstrated clinical stabilization during the follow-up period (from 2 years to 16 years) [2]. However, less than 20% of adults are diagnosed and/or treated by psychiatrists [3]. Indeed, most adults with ADHD received a diagnosis during childhood, but only some of them are currently in contact with psychiatric outpatient services and have access to treatment (e.g., during the transition from child to adult mental health community services). On the other hand, some patients receive a new diagnosis of ADHD during adulthood [4]. It should be highlighted that the management of ADHD in adulthood is essential to prevent poor clinical and functional outcomes associated with this condition, even if comorbidities are treated [5].

The etiology of ADHD is complex, involving multiple biological and environmental factors [6,7]. However, recent evidence has converged to suggest an impairment in catecholamine neurotransmission (especially of dopamine (DA)) in different brain areas of patients with ADHD [8]. Indeed, available treatments for adults include psychostimulants (e.g., methylphenidate (MPH) and amphetamines), noradrenergic compounds (e.g., atomoxetine (ATX) and α2-agonists), and antidepressants increasing DA and noradrenaline (NA) concentrations in the brain, especially in the prefrontal cortex (PFC) [9,10]. Even though pharmacotherapy is recommended in clinical guidelines [11], the efficacy and safety of medications for adult ADHD remain object of debate [12]. According to a relatively recent meta-analysis [13], MPH, amphetamine salts, and bupropion provided better results than a placebo in a global sample of 5000 adults. Of note, considering all included outcomes, amphetamine salts appeared to be the most efficacious compounds, with a tolerability comparable to MPH and an acceptability superior than the placebo. Another meta-analysis, including a total of 1991 adult patients with ADHD, showed that the pooled effect size of all pharmacological treatments versus a placebo was in the medium to high range, although stimulants had higher effect sizes than non-stimulant compounds [14]. Overall, randomized controlled clinical trials (RCT) indicate that MPH and amphetamines are a reliable medication option for the treatment of adult ADHD [9]. According to an analysis of 19 trials regarding adults with ADHD [15], the number needed to treat (i.e., a measurement of the effect of a therapy by estimating the number of patients that need to be treated in order to prevent additional clinical worsening) ranged from two to three for long-acting psychostimulants, from two to four for short-acting psychostimulants, and from two to five for non-psychostimulant compounds. In addition to pharmacotherapy, preliminary data show the effectiveness of neuromodulation techniques (e.g., transcranial magnetic stimulation (TMS) or psychosocial treatments (especially cognitive-behavioral therapy) in improving the symptoms of patients affected by ADHD [16]. However, the treatment response to pharmacotherapy is variable depending on different factors [17]. Some clinical variables, including age at onset, duration of illness, ADHD presentations, having psychiatric (substance use disorders, major depressive disorder, and personality disorder) and/or neurological (epilepsy and headache episodes) comorbidities, and adherence to treatment, were proposed as possible clinical predictors of treatment response [18,19]. On the other hand, even though no biomarkers were identified as a diagnostic tool for ADHD [20], some authors tried to detect biological predictors of treatment response in adults affected by ADHD [21]. In this regard, Scassellati et al. (2012) [22] reported that norepinephrine, 3-methoxy-4-hydroxyphenylethylene glycol, monoamine oxidase, b-phenylethylamine, and cortisol levels may be associated with treatment efficacy. However, a subsequent review by Retz and Retz-Junginger (2014) [18] failed to identify the biological markers of treatment response for adult ADHD. Even though the neuroimaging markers of response to MPH suggested for children included DA transporter status, size of medial PFC, and corpus callosum, these findings were not replicated for adults [6]. On the other hand, the prediction of treatment response may be important for at least two reasons. First of all, the identification of predictors of treatment response is important when facing extremely vulnerable patients such as very young subjects who can complicate the course of the disorder with substance misuse or suicidal behavior [13,23,24,25]. Second, the specific clinical and biological profile of the patients could help clinicians in the choice of the best pharmacological treatment in the context of precision medicine [26,27].

To date, no recent systematic review on the potential clinical and biological predictors of treatment response in adult patients with ADHD has been published. In the light of these considerations, we aimed to systematically review the available data in the literature about the clinical and neurobiological factors that may contribute to predicting the response in these patients.

## 2. Methods

This systematic review was performed according to the Preferred Reporting Items for Systematic Reviews and Meta-Analyses (PRISMA) guidelines [28]. The study protocol was registered in the International Prospective Register of Systematic Reviews (PROSPERO—registration number: CRD42022298758).

A search in the main electronic databases (PubMed, Embase, Scopus, and PsycINFO) was performed using the keywords “(treatment response) AND (adult ADHD)”. Two independent researchers found papers with available abstracts and full-text written in English language from the beginning to May 3, 2022. Firstly, papers were selected according to the pertinence of the title and the information reported in the abstract, then a second selection was conducted after careful reading of the methods and results in the full text.

Inclusion criteria were: (1) age of patients ≥ 18 years; (2) diagnosis of adult ADHD; (3) evaluation of the association between a specific parameter and treatment response in adult ADHD as main topic; (4) clear definition of treatment response using criteria widely accepted in the literature (e.g., reduction in rating scale scores of ≥25% for partial response or ≥50% for full response); (5) clear definition of the psychometric scale used to assess treatment response; (6) English language.

Exclusion criteria consisted of: (1) reviews, meta-analyses, letters to editors, commentaries, comments, case reports and case studies, pooled analyses, and study protocols; (2) animal studies; (3) single-dose pharmacotherapy to evaluate the treatment response; (4) absence of both pre- and post-treatment values of the marker under study. The search strategy and the inclusion and exclusion criteria followed PRISMA guidelines [29].

The two researchers subsequently checked and extracted the following relevant data from included articles: 

Author and title, publication year, primary and secondary outcomes, patients’ characteristics (mean age, gender, and drug free/naive status), trial design, sample size, presence of a control group, tools and criteria used for defining the treatment response, information about ADHD treatment (type, dosage, frequency and duration), type of marker under study, and presence of pre- and post-treatment marker values. If data were not reported in the selected papers, the corresponding author was contacted to obtain further information. Global quality rating was performed according to the criteria proposed by Armijo-Olivo et al. (2012) [30], and the effect sizes were calculated as Cohen’s d.

## 3. Results

The search provided a total of 3855 results. Among these, 1123 were removed as duplicates. In total, 2589 articles were then excluded for the following reasons: 1706 concerned a different topic, 694 were reviews or meta-analyses, 96 were studies conducted on animals, 36 were commentaries or case reports or letters, 30 involved the pediatric population, and 27 were not written in English language. Then, 143 manuscripts were assessed for eligibility, and 121 were discarded because biomarker values for pre- and post-treatment were not reported (*n* = 74) or because the authors did not report a clear definition of treatment response. Finally, 22 articles were included as they satisfied the inclusion criteria (Figure 1).

### 3.1. Genetic Markers

Five pharmacogenomics studies have been conducted to explain the variability of treatment response in individuals with ADHD. Most studies involved genes associated with dopaminergic, serotoninergic, and noradrenergic signaling (Table 1).

**Table 1 jpm-12-01742-t001:** Summary of the studies about genetic markers of treatment response in adult ADHD.

Reference	Markers	Design	*N*	HC	Sex (% Male)	Mean Age ± SD	Drug-Free/Naive	Treatment	Treatment Duration	Treatment Response	Results	ES (d)	Quality of the Study *
Da Silva et al., 2018 [31]	SNARE complex-related genes	Prospective	272	/	55.9%	33.95 ± 10.56	Free	MPH	≥30 days	↓ ≥30% SNAP-IV ↓ CGI ≤ 2 points	SYT1-rs2251214 associated with: (1) short-term response (*p* = 0.006); (2) treatment persistence (*p* = 0.002)	(1) 0.478 (2) 0.291	1
Hegvik et al., 2016 [32]	*ADRA2A*	Observational	564	/	48.0%	34.11 ± 10.0	/	MPH	variable	Options: *- Very good* *- Good* *- Has had effect,* *but discontinued due to side effects* in 2 questionnaires **	rs1800544 in ADRA2A (*p* = 0.033) associated with treatment response	0.320	2
Contini et al., 2012 [33]	*SLC6A4* *HTR1B* *TPH2* *DBH* *DRD4* *COMT* *SNAP25*	Observational	136	7	70.0%	35.0 ± 11.0	Free	MPH	30 days	↓ SNAP-IV	NS	N/A	1
Contini et al., 2011 [34]	*ADRA2A*	Observational	165	/	54.5%	35.0 ± 11.0	/	MPH	30 days	↓ ≥30% SNAP-IV ↓ CGI ≤ 2 points	Three ADRA2A polymorphisms not associated with treatment response (*p* = 0.55, *p* = 0.34, *p* = 0.73 respectively)	0.159, 0.237, and 0.083, respectively	1
Mick et al., 2006 [35]	*DAT1*	RCT	66	40	58.0%	36.9 ± 9.1 35.1 ± 6.9 33.3 ± 6.7	/	MPH	6 weeks	CGI: much/very much improved + ↓ ≥30% AISRS	DAT1 VNTR not associated with treatment response (*p* = 0.90)	0.041	1

Legend: ADRA2A = gene encoding the alpha-2-adrenergic receptor 2A; AISRS = Adult ADHD Investigator Symptom Rating Scale; CGI = Clinical Global Impression scale; d = Cohen’s d; DAT1 = gene encoding dopamine transporter 1; COMT = gene encoding catechol-O-methyltransferase; DBH = gene encoding dopamine β-hydroxylase; DRD4 = gene encoding dopamine D4 receptor; ES = effect size; HC = healthy controls; HTR1B = gene encoding serotonin receptor 1B; MPH = methylphenidate; N/A = not applicable; NS = not significant; *p* = *p* value; RCT = randomized controlled trial; rs = reference Single Nucleotide Polymorphism; SD = standard deviation; SLC6A4 = gene encoding serotonin transporter; SNAP25 = gene encoding synaptosomal-associated protein 25 kDa; SNAP-IV = Swanson, Nolan, and Pelham Rating Scale, version 4; SNARE = soluble N-ethylmaleimide-sensitive factor attachment protein receptors; SYT1 = gene encoding regulatory protein synaptotagmin-1 (SytI); TPH2 = gene encoding tryptophan hydroxylase-2; VNTR = variable tandem repeats; * Armijo-Olivo et al., 2012 [30]; ** Johansson et al., 2008 [36]; Halmoy et al., 2009 [37]; ↓ = reduction.

Da Silva et al. [31] aimed to evaluate the role of genetic variants of neurotransmitter exocytosis-related genes in predicting an immediate-release MPH (IR-MPH) clinical response in a sample of adults with ADHD. In particular, the authors focused on genes codifying for proteins belonging to the SNARE complex (a large protein family that mediates the fusion of vesicles to the target membrane): Syntaxin-1A protein (STX1A, rs2228607), the vesicle-associated membrane protein 2 (VAMP2, 26bp Ins/Del), and the associated regulatory protein synaptotagmin 1 (SYT1, rs1880867 and rs2251214). The original sample consisted of 433 individuals, of which 272 had received MPH for at least 30 days. The main outcomes consisted of a change in the scores of the Portuguese version of the Swanson, Nolan, and Pelham Rating Scale, version 4 (SNAP-IV) and of the Clinical Global Impression-Improvement (CGI-I) scale. Treatment response was evaluated considering the improvement of rating scale scores from baseline to endpoint. Treatment continuation was also assessed in a 7-year follow-up study (long-term protocol). The authors found that only SYT1-rs2251214 was associated with both short-term response to MPH and the amelioration of inattention and oppositional defiant disorder symptoms. Moreover, the presence of SYT1-rs2251214 was found to be associated with a longer duration of treatment with MPH. 

Two studies found mixed results with regard to the adrenoceptor alpha 2A (ADRA2A) gene. Alpha-2-adrenergic receptors are located in the pre- and post-synapses of both the central and peripheral nervous systems. In a study by Hegvik et al. [32], the response to MPH was assessed in 564 adult ADHD patients, and 20 genetic variants were studied in relation to treatment response (selected on the basis of the existing literature). The patients’ response to MPH was assessed by two questionnaires that were filled out by the clinicians. Rs1800544 was found to be the only polymorphism associated with MPH response, although this finding was not confirmed by the meta-analysis grouping the results of the previous studies [32]. In the second study, Contini et al. [34] evaluated the association between three ADRA2A polymorphisms (rs1800544, rs1800545, and rs553668) and the clinical response to MPH in a sample of 165 adults with ADHD. Treatment response to MPH was evaluated by the SNAP-IV and the Clinical Global Impression-Severity (CGI-S) scales. The rating scales were administered at baseline and after one month of treatment. The authors found no significant differences between MPH responders and non-responders in any ADRA2A polymorphisms. 

Another study by Contini et al. [33] investigated the impact of the variants of the following genes in the response to MPH: serotonin transporter (5HTT/SLC6A4), 5-Hydroxytryptamine Receptor 1B (HTR1B), tryptophan hydroxylase-2 (TPH2), dopamine beta-hydroxylase (DBH), catechol-O-methyltransferase (COMT), and synaptosomal associated protein 25 (SNAP25), dopamine receptor D4 (DRD4). The effectiveness of MPH treatment was assessed by the SNAP-IV and the CGI-S scales, administered at baseline and after one month of treatment. No significant differences between MPH responders (*n* = 136) and non-responders (*n* = 28) in any variants of the selected genes were found. 

Finally, a double-blind, 6-week, placebo-controlled trial by Mick et al. [35] had the objective of investigating whether dopamine transporter (DAT1) genetic variants (10-bp variable number of tandem repeats (VNTR) in the 3′-untranslated region (UTR), i.e., homozygous for the 10-repeat DAT1 allele versus heterozygous 9/10 or homozygous 9-repeat) might modulate the effectiveness of MPH in 285 adults with ADHD. The original sample was randomized to IR-MPH or placebo and to osmotic release oral system (OROS)-MPH or placebo. Patients were randomized to MPH or placebo at a ratio of 2.5:1. The Adult ADHD Investigator System Report Scale (AISRS) and CGI scale were used to evaluate the treatment response. DAT1 variants were investigated in 66 (32 OROS- and 34 immediate-release) MPH subjects and 40 placebo subjects. The findings of this study failed to support the role of DAT1 VNTR in the 3′ UTR in the treatment response to MPH. 

#### Conclusions

Only SYT1-rs2251214 may be related to clinical amelioration by treatment. Instead, the other studies did not identify any genetic variants associated with treatment response in adults affected by ADHD.

### 3.2. Neuroimaging Markers

The majority of studies focused on task-based functional neuroimaging. Among different neuroimaging biomarkers, the striatum was the most extensively studied region (Table 2).

**Table 2 jpm-12-01742-t002:** Summary of the studies about neuroimaging markers of treatment response in adult ADHD.

Reference	Markers	Brain Imaging Technique	Design	*N*	HC	Sex (% Male)	Mean Age ± SD	Drug-Free/Naive	Treatment	Treatment Duration	Treatment Response	Results	ES (d)	Quality of the Study *
Chang et al., 2021 [38]	Volume of left putamen and precuneus	fMRI (T1-weighted images)	Machine learning comparison: good responders (*n* = 63) versus poor responders (*n* = 16)	79	/	54.0%	18.0 ± 10.0	Naive	MPH	1 month	CGI: much/very much improved	Poor responders: (1) smaller regional GM volumes in the left putamen (*p* = 0.010); (2) higher volumes in the right (*p* = 0.025); (3) left (*p* = 0.031) precuneus	(1) 0.651 (2) 0.655 (3) 0.548	2
Sugimoto et al., 2021 [39]	Regional brain activity	NIRS	Prospective interventional: near-infrared spectroscopy examinations during a go/no-go task pre- and post-treatment	31	/	61.3%	31.2 ± 8.6	Free	ATX	8 weeks	↓ CAARS; ADHD test program	↑ PFC activity associated with severity of clinical symptoms before treatment in the group of non-responders	N/A	2
Sethi et al., 2018 [40]	Substantia nigra/ventral tegmental area, ventral striatum	fMRI (T2-weighted images)	RCT	30	30	63.3%	33.7 ± 9.5	/	D-AMP/MPH	≥2 months (suspended 2 days before the test)	N/A	ADHD patients in treatment showed a change in signalling of left ventral striatum (*p* = 0.042)	1.67	1
Fan et al., 2017 [41]	Inhibitory control visual processing (Stroop fMRI and CANTAB)	fMRI	RCT	24	12	71.4%	28.9 ± 7.8	Naive	ATX	8 weeks	↓ CGI ↓ ASRS	Pre-treatment anterior cingulate activation improvement of clinical symptoms (*p* < 0.05)	N/A	2
Volkow et al., 2012 [42]	Dopamine release changes (PET)	PET scans	Prospective, single-blind	20	/	40.0%	32.0 ± 6.0	Naive	MPH	12 months	↓ CAARS	Responders: ↑ DA in ventral striatum	N/A	2
Bush et al., 2008 [43]	Increased fMRI activation in the daMCC and other frontoparietal regions involved in attention (MSIT)	fMRI	RCT	11	10	63.6%	29.5 ± 5.9	Free	MPH	6 weeks	↓ ADHD ISRS ↓ CGI 1 or 2 points	Degree of daMCC activation was related to treatment response	1.084	2
La Fougère et al., 2006 [44]	Striatal DAT captation	SPECT	CT	22	14	50.0%	37.8 ± 11.0	Free	MPH	10 weeks	↓ CGI	Poor response: pre-treatment ↓ striatal DAT binding (*p* = 0.04)	2.334	2
Krause et al., 2005 [45]	DAT availability	SPECT	CT, single-blind	18	14	55.6%	39.5 ± 11.1	Naive	MPH	10 weeks	↓ CGI	Among the group of responders, all patients had high DAT availability prior to therapy; all except one of non-responders presented a low DAT availability before treatment	2.491	2
Schweitzer et al., 2003 [46]	- Posterior cerebellum - Precentral gyri - Left caudate nucleus - Right claustrum	PET	Prospective, open-label	10	/	100.0%	31.5 ± 8.2	/	MPH	3 weeks	↓ CGI ↓ ADHD-RS	Change in clinical symptoms after the treatment was negatively correlated with rCBF increases in the midbrain, cerebellar vermis, and the precentral and middle frontal gyri in the off-MPH condition	N/A	1

Legend: ADHD ISRS = Adult ADHD Investigator Symptom Report Scale; ADHD-RS = ADHD Rating Scale; ASRS = Adult ADHD Self-Report Scale; ATX = atomoxetine; CAARS = Conners’ Adult ADHD Rating Scales; CANTAB = Cambridge Neuropsychological Test Automated Battery; CGI = Clinical Global Impression scale; CGI-I = Clinical Global Impression-Improvement subscale; CT = clinical trial; d = Cohen’s d; DA = dopamine; daMCC = dorsal anterior midcingulate cortex; D-AMP = dextroamphetamine; DAT = dopamine transporter; ES = effect size; fMRI = functional magnetic resonance imaging; GM = gray matter; HC = healthy controls; MPH = methylphenidate; MSIT = Multi-Source Interference Task; N/A = not applicable; *p* = *p* value; PET = positron emission tomography; PFC = prefrontal cortex; rCBF = regional cerebral blood flow; RCT = randomized controlled trial; SD = standard deviation; NIRS = near-infrared spectroscopy; SPECT = single photon emission computed tomography; * Armijo-Olivo et al., 2012 [30]; ↓ = reduction; ↑ = rise.

#### 3.2.1. Structural Neuroimaging Studies

A study by Chang et al. [38] included 79 drug-naive individuals with ADHD, who were subsequently treated with MPH for at least one month. The authors aimed to identify the neuroanatomical features that could differentiate between good and poor responders to pharmacological treatment. Particularly, the research group focused on the striatum and other regions of the default-mode network (DMN) (brain circuitry influencing MPH effects and the response to treatment) including the posterior cingulate/precuneus, medial PFC, and lateral inferior parietal cortex. According to CGI-I, 63 individuals were allocated in the good responder group, while 16 were in the poor responder group. The authors found that poor responders had smaller regional volumes of the left putamen as well as larger precuneus volumes compared to good responders at baseline.

#### 3.2.2. Dopamine Levels and Regional Cerebral Blood Flow

A prospective study by Volkow et al. [42] included 20 treatment-naive adults with ADHD before treatment initiation and after one year of MPH. In particular, the authors assessed whether the magnitude of DA increases, induced by acute intravenous administration of MPH, could predict the clinical response to long-term treatment with oral MPH. Participants twice underwent positron emission tomography (PET) after the acute administration of a placebo or intravenous MPH: one at baseline and one after 12 months of treatment with oral MPH. Effectiveness of treatment was evaluated using the Conners’ Adult ADHD Rating Scale (CAARS) as follows: CAARS-A for inattention, CAARS-B for hyperactivity, and CAARS-H for global symptoms. The rating scale scores were used to assess the correlations between clinical changes and the magnitude of the brain DA increases induced by intravenous MPH. The authors found that intravenous MPH was significantly associated with an increase in dopamine in the striatum (rated as decreases in D2/D3 receptor availability). In the ventral striatum, the magnitude of DA increase was associated with the amelioration of inattention with long-term oral MPH treatment. Moreover, statistical parametric mapping showed that DA increases in prefrontal and temporal cortices due to intravenous MPH were related to the improvement of inattention. 

Schweitzer et al. [46] aimed to define the anatomy of regional cerebral blood flow (rCBF) responses associated with MPH and to identify resting patterns of rCBF that might predict response to MPH among 10 untreated adult males with ADHD. The measures of rCBF using PET were acquired before treatment (or off-MPH condition) and after a 3-week period of treatment with MPH (or on-MPH condition). Compared with the on-MPH condition, the off-MPH one was associated with bilateral relative increases of rCBF in the precentral gyri, left caudate nucleus, and right claustrum. The on-MPH condition was associated with relative increases of rCBF in the cerebellar vermis and in a site lateral to the vermis in the right cerebellar hemisphere. The change in clinical symptoms after the 3-week period of treatment with MPH (evaluated through CGI and ADHD rating scale (ADHD-RS) scores) was negatively correlated with rCBF increases in the midbrain, cerebellar vermis, and the precentral and middle frontal gyri in the off-MPH condition.

#### 3.2.3. Dopamine Transporters

La Fougère et al. [44] aimed to evaluate whether the degree of dopamine transporter (DAT) binding in the striatum could predict the response to a 10-week treatment with MPH in a sample of 22 drug-naive patients with ADHD. Among patients with high striatum DAT binding (*n* = 17) prior to therapy, all except one individual responded well to MPH therapy (as estimated by CGI before and after MPH medication), while none of the ADHD patients with low DAT levels (*n* = 5) showed a significant improvement of ADHD symptoms. Similarly, Krause and collaborators [45] compared striatum DAT availability before starting therapy in MPH responders (*n* = 12) and non-responders (*n* = 6) to 10-week treatment. The patients’ response to MPH was assessed by CGI. Among the group of responders, all patients had high DAT availability prior to therapy, while all except one of non-responders presented a low DAT availability before treatment. Furthermore, a significant negative correlation between global clinical improvement and striatal DAT availability was reported.

#### 3.2.4. Task-Based Functional Neuroimaging

A recent study by Sugimoto et al. [39] evaluated 31 adult patients with ADHD who underwent near-infrared spectroscopy examinations during a go/no-go task, both before and after 8-week atomoxetine (ATX) administration. Near-infrared spectroscopy was acquired during a 10-minute computerized, visual-response, inhibition task. In this go/no-go task, the non-target stimulus A and a target stimulus B, which closely resembled A, were randomly presented. The participant was asked to press the space key using their index finger as quickly as possible when A was presented (response) and to refrain from pressing the space key when B was presented (response inhibition). CAARS was administered to assess the clinical symptoms of the patients, both at baseline and 8 weeks after ATX treatment. The authors found a positive correlation between right dorsolateral prefrontal cortex activity and CAARS scores. Moreover, when participants were divided into responders and non-responders, a positive correlation was observed between prefrontal cortex activity and clinical symptoms only in the group of non-responders. 

Sethi and co-authors [40] tested 30 patients with ADHD and 30 matched controls by a reinforcement-learning task, which demonstrated to be reliable in assessing the effects of novelty on reward-related behavior. Each participant completed the task during functional magnetic resonance imaging (fMRI) on two separate occasions, once after taking stimulant medication (on-medication) and the other after a placebo (off-medication). The patients without medication showed an impaired task performance and a greater selection of novel options. Moreover, persistence in selecting novel options predicted impaired task performance. These deficits were accompanied by a significantly lower learning rate and heightened novelty signaling within the substantia nigra/ventral tegmental area. Compared to the controls, the ADHD patients under stimulant medication improved their overall task performance, increased their reward-learning rates, and enhanced their ability to differentiate optimal from non-optimal novel choices. The medication also reduced novelty signaling in substantia nigra/ventral tegmental area. For this reason, this parameter could be a potential predictor of response to treatment.

Using the counting Stroop task combined with fMRI and the Cambridge Neuropsychological Test Automated Battery (CANTAB), an 8-week placebo-controlled study by Fan and co-authors [41] investigated the treatment-related changes in inhibitory control and visual processing in 24 drug-naive adults with ADHD. Participants were randomly assigned to either ATX or placebo. Rapid visual information processing (RVP) and delayed matching to sample (DMS) were chosen from the CANTAB to assess inhibitory control and visual processing, respectively. The RVP test is designed to examine sustained attention capacity and inhibitory control, whilst DMS assesses the participants’ visual processing and short-term visual memory, which are cognitive abilities used to remember the visual features of a complex target stimulus in a four-choice delayed recognition. The authors found that ATX decreased activations in the right inferior frontal gyrus and anterior cingulate cortex, while improving inhibitory control. Moreover, ATX increased activation in the left precuneus, contemporary to an improvement in the mean latency of correct responses. Finally, anterior cingulate activation in pre-treatment predicted the effectiveness of ATX. 

A RCT by Bush et al. [43] included 21 adults with ADHD who were randomized to receive a 6-week treatment with MPH OROS (*n* = 11) or a placebo (*n* = 10). The study was finalized to determine whether MPH favored the activation of the dorsal anterior midcingulate cortex (daMCC) and other fronto-parietal regions involved in enhancing attention during the Multi-Source Interference Task (MSIT). Severity and clinical response were assessed using the AISRS and CGI scales. Compared with the placebo, treatment with MPH significantly increased daMCC activation during MSIT. Moreover, the degree of daMCC activation was related to treatment response. In the treatment group, 71% (5 of 7) of the responders (compared to 25% of non-responders) showed an increase from baseline to endpoint in daMCC activity. 

#### 3.2.5. Conclusions

Some imaging markers related to DMN-striatum connectivity might have a significant role to predict treatment response in adult ADHD. 

### 3.3. Neurophysiological Markers

A study on the antisaccade task (AST) and another one on executive functions as possible predictors of treatment response among adults with ADHD were published (Table 3).

**Table 3 jpm-12-01742-t003:** Summary of the studies about neurophysiological markers of treatment response in adult ADHD.

Reference	Markers	Design	*N*	HC	Sex (% Male)	Mean Age ± SD	Drug-Free/Naive	Treatment	Treatment Duration	Treatment Response	Results	ES (d)	Quality of the Study *
Duval et al., 2021 [47]	Antisaccade task performance	Prospective	97	50	49.5%	35.1 ± 9.5	Naive	MPH	6 months	↓ ASRS ↓ CGI	Low percentage of direction errors after the first MPH dose predicted remission after 6 months of pharmacotherapy (*p* = 0.001)	0.973	2
Biederman et al., 2011 [48]	Attentional networks (Attention Network Test); Inhibitory control (Stop Signal Test)	Cross-sectional RCT	87	146	40.0%	34.7 ± 9.2	/	MPH	6 weeks	CGI: much/very much improved (≤2) + ↓ ≥30% AISRS	EFDs: (1) do not moderate the response to MPH (*p* = 0.35); (2) are not associated with response to MPH (0.794)	(1) N/A (2) 0.255	1

Legend: AISRS = Adult ADHD Investigator Symptom Rating Scale; ASRS = Adult ADHD Self-Report Scale; CGI = Clinical Global Impression scale; d = Cohen’s d; EFDs = executive function deficits; ES = effect size; MPH = methylphenidate; N/A = not applicable; *p* = *p* value; RCT = randomized clinical trial; SRTs = saccade reaction times; * Armijo-Olivo et al., 2012 [30]; ↓ = reduction.

Duval et al. [47] investigated whether AST performance after the first MPH dose may be associated with subsequent clinical amelioration in adults with ADHD. Saccadic eye movements were extensively used as a research tool to investigate the working functions of the brain. In particular, the AST evaluates the mechanisms of voluntary saccade control, requiring the inhibition of a reflexive eye movement towards a visual target and a volitional saccade directed away from the target to the (unmarked) mirror-symmetrical location. Such a response involves a complex neural circuitry including regions within the occipital, parietal, and frontal cortices, superior colliculus, thalamus, striatum/basal ganglia, brainstem reticular formation, and cerebellum. Ninety-seven drug-naive ADHD adults were included. The AST parameters, as well as the ASRS and CGI scores, were collected at baseline, after the first MPH dose (2 weeks later), and 6 months after chronic MPH treatment. The results were compared with those of 50 HC. Acute and chronic MPH administration resulted in normalization of the AST performances. Furthermore, a low percentage of direction errors after the first MPH dose predicted remission (i.e., after 6 months of pharmacotherapy, final Adult ADHD Self-Report Scale (ASRS) score ≤ 9). In addition, a 6-week, placebo-controlled trial by Biederman et al. [48] had the objective of evaluating the association between executive function deficits (EFDs) and response to MPH in adults affected by ADHD. The patients were randomized (1:1) to receive OROS-MPH or placebo. Subjects with available measures of EF (OROS-MPH *n* = 40; placebo *n* = 47) were included. The Behavior Rating Inventory of Executive Function for Adults (BRIEF-A) was administered at baseline and endpoint (6 weeks) to assess behavioral manifestations of EFDs. Overall, 40% of the included subjects had EFDs at baseline. The authors failed to find an association between EFDs at baseline and subsequent treatment response to MPH.

#### Conclusions

To date, the paucity of studies does not allow the authors to draw definitive conclusions on the possible role of neurophysiological markers to predict treatment response for ADHD. 

### 3.4. Electrophysiological Markers

As shown in Table 4, seven electrophysiological parameters were investigated as predictors of treatment response.

**Table 4 jpm-12-01742-t004:** Summary of the studies about electrophysiological markers of treatment response in adult ADHD.

Reference	Markers	Design	*N*	HC	Sex (% Male)	Mean Age ± SD	Drug-Free/Naive	Treatment	Treatment Duration	Treatment Response	Results	ES (d)	Quality of the Study *
Alyagon et al., 2020 [49]	Alpha and low-gamma power	Semi-blinded RCT	15	AC: 14 Sham treatment: 14	13.3%	26.6 ± 0.7	Naive	rTMS	3 weeks	↓ ≥25% CAARS (total and subscales); ↓ BAARS-IV; ↓ BRIEF-A; ↓ BDI	Treatment response associated with: (1)↓ α activity (*p* = 0.035) (2) ↑ low-γ (*p* = 0.012) (3) ↓ α + ↑ low-γ (*p* = 0.0001)	(1) 1.352 (2) 2.200 (3) 3.695	1
Werner et al., 2020 [50]	Retinal background noise	Longitudinal	20	21	55.0%	30.5 ± 10.0	Naive	MPH	7 weeks	↓ CAARS	↓ retinal background noise at follow-up after treatment in ADHD patients (*p* = 0.035); no changes in HC	0.847	2
Strauss et al., 2020 [51]	Brain arousal instability during resting-state EEG (Vigilance Algorithm Leipzig (VIGALL 2.1))	Open-label	28	/	28.6%	36.6 ± 11.6	Free	MPH	4 weeks	↓ CAARS	Arousal stability at baseline predicted MPH response (*p* = 0.027)	N/A	2
Cooper et al., 2014 [52]	VLF-EEG oscillation	Longitudinal case-control	17	34	100.0%	28.7 ± 7.7	Free	MPH	Mean: 9.4 months	↑ CPT OX	VLF-EEG activity and omission errors reduced in cases after treatment to the same level as healthy controls	N/A	2
Leuchter et al., 2014 [53]	qEEG absolute and relative power, cordance	RCT	14	15	N/A	18–30	Free	ATX	12 weeks	↓ CAARS; ↑ AAQOL-29	Responders and non-responders to ATX differed significantly in week 1 theta band left tempoparietal cordance, with atomoxetine responders showing the lowest and non-responders the highest values (*p* = 0.015)	1.503	2
Skirrow et al., 2015 [54]	Frontal theta activity	Open-label	21	36	100.0%	30.0 ± 10.4	Free	MPH	Mean: 3.5 months	↑ CPT OX/SART	Normalisation of frontal theta activity pattern after treatment (*p* = 0.02)	N/A	2

Legend: AAQOL-29 = adult ADHD quality of life questionnaire; AC = active control; ATX = atomoxetine; BAARS-IV = Barkley Adult ADHD Rating Scale; BDI = Beck’s Depression Inventory; BRIEF-A = Behavior Rating Inventory of Executive Function for Adults; CAARS = Conners’ Adult ADHD Rating Scales; CPT OX = continuous performance test with flankers; d = Cohen’s d; EEG = electroencephalogram; ES = effect size; MPH = methylphenidate; N/A = not applicable; *p* = *p* value; qEEG = quantitative EEG; RCT = randomized clinical trial; rTMS = repetitive transcranial magnetic stimulation; SART = sustained attention to response task; VLF = very low frequency; * Armijo-Olivo et al., 2012 [30]. ↓ = reduction; ↑ = rise.

On the basis that elevated neuronal background noise (or non-stimulus-driven neuronal activity) was reported as a pathophysiological correlate of ADHD, Werner et al. [50] assessed retinal background noise with the pattern-electroretinogram (PERG) in 20 drug-naive adults with ADHD before and after treatment with MPH (at an average of 7 weeks following the first visit) and in 21 control subjects (at baseline and approximately 3 weeks after the first consultation). Background noise was identified using the Fourier magnitude at frequencies adjacent to the stimulus-response frequency of 12.5 Hz. Clinical symptoms were rated in all participants using the CAARS and the Wender Utah Rating Scale (WURS). At baseline, the background noise in the untreated ADHD patients was significantly higher than in the healthy controls. ADHD patients showed a significant reduction in the background noise at the second visit during treatment compared to the first visit before therapy. On the contrary, healthy controls did not show significant changes in the background noise between the baseline and the endpoint. Other three studies were carried out using MPH as the main treatment. In the first one, Strauss et al. [51] investigated whether baseline brain arousal instability during resting-state EEG could predict response to 4-week MPH treatment. Indeed, according to the arousal regulation model of ADHD and affective disorders, hyperactivity may be interpreted as an auto-regulatory attempt to stabilize brain arousal in analogy to the behavioral pattern elicited by overly tired children. EEGs were recorded at baseline and 4 weeks after the initiation of MPH treatment. CAARS scores were used to define treatment response. The sample consisted of 28 drug-free adult patients with ADHD. Among these, 8 and 20 subjects were, respectively, categorized as responders and non-responders to MPH. The group of responders, compared to the group of non-responders, had less arousal stability during the 15-min EEG at baseline. Moreover, 4 weeks after initiation of MPH, these differences in arousal stability between responders and non-responders were no longer present, indicating a normalization of arousal stability by MPH treatment. A less stable arousal regulation at baseline predicted, therefore, a higher chance of response to MPH. The second study by Cooper et al. [52] investigated the case-control differences in very low-frequency (VLF) EEG (<0.01 Hz) activity in adult samples and the effects of MPH during a cued continuous performance task (CPT-OX task). VLF-EEG was proposed as the potential endophenotype of ADHD, reflecting gross cortical excitability. In this study, 41 untreated adults with ADHD and 47 controls were included: 21 cases were followed up after MPH initiation, together with 38 controls. The authors found that cases had an enhanced frontal and parietal VLF-EEG activity and more omission errors (failures to answer when a response was required) than controls. In the whole sample, an increased parietal VLF-EEG activity correlated with the number of omission errors. In the post-treatment phase, a time by group interaction emerged. Indeed, VLF-EEG activity and omission errors reduced in cases to the same level as controls, such that VLF-EEG pattern can be a potential predictor of treatment response. Skirrow et al. [54] investigated, among 41 adults with ADHD and 48 controls, cortical activation during different conditions including a resting state, a cued CPT-OX, and the sustained attention to response task (SART). Moreover, the study had the objective to investigate the effects of MPH in a subsample of 21 ADHD cases. Mean time of follow-up was 3.5 months. The authors found that control participants showed a task-related increase in theta activity when engaged in cognitive tasks, mainly in the frontal and parietal regions, which was absent in adults with ADHD. Nevertheless, treatment with MPH resulted in normalization of the EEG theta activity in patients, such that this parameter can be a potential predictor of treatment response. 

Leuchter et al. [53] investigated thalamocortical oscillatory activity using quantitative EEG (qEEG absolute power, relative power, and cordance) as a possible biomarker of ADHD, as well as a potential predictor of a response to ATX. A number of qEEG studies showed abnormal patterns of both low and high frequency neuronal oscillatory activity in ADHD, as a result of a deficit in the integrative or inhibitory processing in the default mode network regulation by the subcortical structures (thalamus and striatum). Absolute and relative power were calculated in four frequency bands, corresponding to delta (0.5–4 Hz), theta (4–8 Hz), alpha (8–13 Hz), and beta (13–20 Hz). qEEG measures were calculated in this randomized, double-blind, placebo-controlled trial from a subset of 29 individuals (ATX, *n* = 14; placebo, *n* = 15) who had useable qEEG recordings. The primary study outcome was the improvement of ADHD symptoms evaluated by the administration of CAARS at baseline, after two weeks of treatment, and at endpoint (12 weeks of treatment). The authors found that left temporoparietal cordance in the theta frequency band after one week of ATX was associated with ADHD symptom improvement and the amelioration of quality of life in subjects receiving the active compound, but not in those treated with the placebo. Specifically, responders and non-responders to ATX differed significantly in week 1 theta band cordance, with atomoxetine responders showing the lowest values and non-responders the highest values.

Alyagon et al. [49] aimed to evaluate whether treating the right prefrontal cortex (rPFC) with multiple sessions of repetitive transcranial magnetic stimulation (rTMS), would improve the clinical symptoms of adults suffering from ADHD. Thus, the authors used EEG in order to characterize electrophysiological alterations induced by treatment and to look for eventual correlations between baseline neuronal activity and clinical response. Forty-three drug-free adults with ADHD were randomized to receive either real treatment (*n* = 20), sham treatment (*n* = 16), or an active control (*n* = 16; off-target stimulation), and they underwent 3 weeks of daily high-frequency (18 Hz) stimulation sessions. During every visit, the participants completed the CAARS, Barkley Adult ADHD Rating Scale (BAARS-IV), Behavioral Rating Inventory for Executive Functioning (BRIEF-A), and Beck Depression Inventory (BDI) scales. Real treatment (final sample, *n* = 15) resulted in a significant improvement of symptoms. Alpha activity negatively correlated with clinical amelioration, while low-gamma power positively correlated with the improvement of symptoms. Furthermore, the ratio of low-gamma power to alpha activity highly correlated with the amelioration of ADHD symptoms.

#### Conclusions

According to heterogeneity among studies, the findings are preliminary and need further confirmation in larger samples. 

## 4. Discussion

As about one-third of adults affected by ADHD show a poor response to the available treatments [55], we aimed to review the original studies about biological predictors of treatment response in adults affected by ADHD. The results from the different studies do not identify robust biological predictors of treatment effectiveness in these patients.

With regards to genetic studies, our systematic review showed that only the single nucleotide polymorphism (SNP) on the SYT1 gene (rs2251214 A-carriers) was associated with clinical amelioration by treatment, specifically in the domains of inattention and hostility [31]. SYT1 is a synaptic vesicle protein that functions as a calcium sensor and is a main regulator of the SNARE complex that plays a key role in the brain by controlling neurotransmitter release [56]. Preliminary evidence from animal studies suggests that variants of the SNARE complex may confer susceptibility to develop ADHD [57]. In this regard, Cupertino et al. [58] reported that the G allele of rs2251214 may be related to an earlier onset of impairment due to ADHD symptoms, higher incidence of behavioral problems, and comorbid antisocial personality disorder. Nevertheless, it is noteworthy that personality disorders may be predictors of poor outcomes in ADHD and limited response to MPH [5,59]. Thus, we cannot completely exclude a shared genetic risk factors among ADHD and personality disorders, which are, in turn, associated with poor outcomes [31]. The other studies included in the present review [32,33,34,35] failed to identify other genetic variants associated with treatment response in adults affected by ADHD, including ADRA2A. Even though ADRA2A is highly expressed in the brain areas involved in ADHD [60], it is likely that larger samples are required to identify an effect of variants of this gene in the prognosis of ADHD [32]. Nevertheless, psychiatric comorbidities as well as environmental factors, combined in different ways, may contribute to the phenotypic heterogeneity of ADHD. Furthermore, some symptoms of ADHD, including hyperactivity and impulsivity, may mitigate, along with age, differently from inattention [61]. In addition, MPH may not be effective against all ADHD symptoms in the same way, and the effect of MPH may change over time as result of alterations in gene expression patterns [34,62]. Consequently, pharmacogenetic studies carried out on children affected by ADHD might not be directly applicable to adults. So far, even though genome-wide association studies offer a powerful means to find the genes involved in different neural activities [33,35], only a small number of loci possibly associated with ADHD have been identified until now [19]. Taken as a whole, more genetic studies are needed, even for genes that were already investigated in relation to ADHD prognosis (e.g., DRD4, SNAP 25, and COMT) [63].

With regard to the topic of the present article, neuroimaging studies have the objective of identifying changes in the different brain regions associated with response to treatment. Even though neuroimaging is an emerging field, this approach might be helpful in the prediction of therapeutic response only in some specific conditions such as mood disorders [64]. On the other hand, the number of this type of study on ADHD patients is very limited. The findings summarized in the present article should be, therefore, interpreted not only as promising but also as very preliminary. Nevertheless, some interesting results emerged from our research. Larger striatum volumes were found to be associated with good MPH response [38]. Larger striatal volumes might account for higher D2 receptor availability locally, resulting in a higher effectiveness of MPH [45]. Nevertheless, DAT status in untreated ADHD individuals could be a useful predictor of response to treatment, since MPH could increase striatal dopamine availability through the binding to DAT [46]. Of note, long-term treatment with MPH results in upregulation of DAT in the striatum, as a result of the neuroplasticity effects induced by such a molecule [42]. It is also noteworthy that, given functional connectivity between the striatum and different brain regions, MPH-associated brain structural and functional changes could also involve other areas that might be related to treatment response as well. MPH, inducing an increase in DA and NA neurotransmission, suppresses DMN activities (involving the PFC, parietal cortex, and precuneus/cingulated cortex), resulting in clinical amelioration of patients with ADHD [38]. Nevertheless, Volkow et al. [38] reported that the enhancement of DA signaling in prefrontal and temporal cortices was related to an improvement of attention when individuals were treated with MPH. Even though it has been hypothesized that prefrontal regions of ADHD patients express low DAT levels [65], MPH also blocks the NA transporters, which in PFC contribute to DA increases [66], as reported in fMRI studies [41]. Furthermore, Sethi et al. [40] reported that treatment with MPH resulted in a significant reduction in signaling in the substantia nigra/ventral tegmental area, which is associated with the amelioration of cognitive performances. In particular, aberrant novelty processing in reward-related decision-making was reported as a component of the impulsive phenotype of ADHD [67]. In this framework, some authors hypothesized an association between D2/D3 receptor density in the substantia nigra/ventral tegmental area, novelty-seeking behavior, and response to MPH [68]. MPH-related changes also include a reduction in the rCBF in some brain areas associated with excessive motor activity and impaired inhibition typical of ADHD [69]. Schweitzer et al. [46] suggested that higher activity of the midbrain, middle frontal gyrus, and posterior cerebellum, which are regions of DA projection, is associated with a lower probability to respond to MPH. As well as MPH, Sugimoto and collaborators [39] found that ATX administration increased DLPFC activity, indicating that this region may be involved in the clinical symptoms of ADHD. Of note, cognitive impairment could be more severe in patients with more intense PFC activity when performing the same task; however, the available literature is contradictory [70]. Nevertheless, the degree of ACC activation before treatment with ATX may be predictive of subsequent clinical response to this compound [40]. Particularly, ATX seems to decrease the activation of ACC during emotional-response inhibition and visual processing, possibly acting on neurotransmission in the precuneus similarly to MPH [38]. 

Some studies investigated EEG-based markers in adult patients with ADHD, based on the assumption of sub-optimal arousal in these subjects. An impaired arousal seems to be involved in ADHD symptoms [71]. Particularly, DMN reflects a pattern of brain activity that is dominant when a subject is at rest and attenuated when performing cognitive tasks. DMN may be, therefore, inadequately attenuated when subjects affected by ADHD perform cognitive tasks [72]. Theta oscillations (4–8 Hz) could be implicated in the attenuating process required for cognitive functioning [73]. Of note, Skirrow et al. [54] reported a significant attenuation in theta activity when adult patients with ADHD were engaged in cognitive tasks. On the other hand, theta cordance from the left tempoparietal region after one week of treatment with ATX was significantly related to long-term clinical amelioration of ADHD symptoms. Particularly, individuals with lower theta cordance values at one week showed greater improvement in all ADHD symptoms than those with higher theta cordance [54]. Nevertheless, the authors suggested that their findings involving cordance in the theta frequency band represent a sort of thalamocortical arrhythmia, possibly reflecting an important mechanism associated with the pathophysiology of the illness as well as the effect of treatment [74]. In addition to theta activity, altered VLF-EEG (0.02–0.2 Hz) activity was proposed as a possible endophenotype of ADHD in adult patients. Particularly, altered VLF fluctuations may reflect a dissociation between the frontal and parietal regions within DMN [75]. Cooper et al. [52] found that MPH may normalize VLF-EEG activity to the same levels of controls, with an improvement of attention. Of note, ADHD is characterized by multiple functional and structural neural network abnormalities, most prominently of the frontal networks [6]. In this regard, Alyagon et al. [49] reported that alpha (8–14.5 Hz) and low-gamma (30–40 Hz) activity had an opposite effect in predicting treatment response in patients with ADHD. Furthermore, ADHD individuals exhibit hypo-activation of the right PFC [76], and, therefore, patients with ADHD with less right and more left alpha PFC activity may respond better to treatment [77]. Finally, other electrophysiological predictors of treatment response including retinal background noise [50] as well as arousal stability [51], although promising, are preliminary and need further confirmation in larger studies. 

Very few studies have investigated cognitive markers as predictors of treatment response. Duval et al. [47] reported normalization of AST performance after acute or chronic administration of MPH. Moreover, a lower percentage of errors after the first dose of MHP is associated with clinical subsequent remission. In line with the hypothesis of frontostriatal hypo-activation in ADHD, MPH may restore voluntary inhibitions on saccades by increasing catecholaminergic activity, particularly in PFC [42]. On the other hand, EFDs do not seem to have an impact on response to MPH [48]. To date, the literature has reported some discrepancies between child and adult findings; in addition, it is not still clear if the acute effectiveness of MPH on cognition persists in the long term. [78]. Further studies are needed to confirm if it is possible to identify neurocognitive predictors of treatment response in adult ADHD patients. 

All the interpretations and the generalizability of the findings from this review should be considered cautiously for different limitations. First of all, most studies were based on a relatively small sample size, with eight of these [39,42,43,45,46,49,50,52,53] including 20 or fewer participants. Second, only a few findings [35,40,41,43,48,49,53] are secondary results from RCT. RCT is considered the “gold standard” for intervention studies, as this design establishes the notion of control, thereby minimizing bias. However, inclusion/exclusion criteria of these types of studies may not exactly reflect the clinical real world. Moreover, in line with our findings, the duration of the treatment in RCT is usually limited to some weeks, which could be an amount of time that is too short, in most cases, to assess a reliable treatment response/remission. Nevertheless, treatment response and remission have been determined in different ways. Currently, there is no absolute consensus on the criteria by which remission and response are defined. Third, studies were conducted applying different techniques, methods, rating scales, and treatment duration. Particularly, some studies of neuroimaging and electrophysiological markers [38,40,42,44,45,53] used the region of interest (ROI) approach. However, the ROI approach is highly vulnerable to type II errors, given that treatment effects in regions other than those driven by an a priori hypothesis might be neglected [6]. In the same way, genetic studies have focused on few variants, leading to the reduced power of their analysis. Nevertheless, high levels of dropout and discontinuation rates, the absence of placebo control groups, and the exploratory nature of some studies may have contributed to high heterogeneity between studies, thus severely affecting the generalizability of the results. Finally, some effect sizes cannot be calculated because of insufficient published data. 

## 5. Conclusions

The systematic research in the literature revealed only a very small number of studies about the topic of this article: the identification of relevant biological predictors of treatment response in adults affected by ADHD. Polymorphisms in SNARE complex genes and markers related to DMN-striatum connectivity might have a significant role in predicting treatment response in adult ADHD. However, small-to-moderate effect sizes may limit their utility as prognostic biomarkers [79]. Even though the results of the present review may be promising for improving diagnostic accuracy, additional research will have to quantify the degree of reliability of the different biological markers. Particularly, machine learning approaches can be useful to confirm the reliability of biological predictors of treatment response [80,81]. In addition, some biomarkers must, probably, still be identified in the light of the complex pathophysiology of ADHD and the differences in the biological and clinical aspects between adults and children [18]. However, studies regarding the long-term prognosis of adult ADHD patients are lacking. Further studies with larger samples, taking into account the three presentations of the disorder (predominantly inattentive, predominantly hyperactive-impulsive, and combination) and other environmental or biological factors are needed to have a complete overview of this topic [5,82].

## Figures and Tables

**Figure 1 jpm-12-01742-f001:**
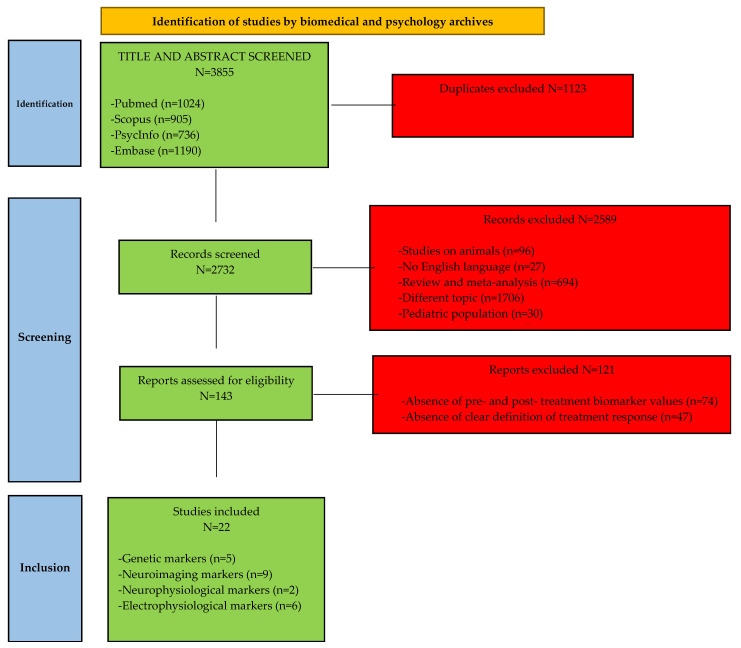
PRISMA flowchart for systematic reviews.

## Data Availability

The datasets generated during and/or analysed during the current study are available from the corresponding author on reasonable request.

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
