# Peer review of "Biological Predictors of Treatment Response in Adult Attention Deficit Hyperactivity Disorder (ADHD): A Systematic Review"

_jpm, 2022, doi:10.3390/jpm12101742_

Round 1

Reviewer 1 Report

This well-written review by Capuzzi et al summarizes findings in biological predictors of adult ADHD treatment responses. Four broad types of biological predictors were surveyed and result summaries of these studies are nicely illustrated. Biomarkers of psychiatric disorders are at a rather early stage, and this review provides references for further investigation in adult ADHD treatment responses.

Major issues:

  1. Line 43: “albeit up to 65% of affected children may continue to manifest symptoms in adulthood”.

According to Sibley et al, only 9.1% of the subjects demonstrated recovery by the study endpoint of 25 years of age. The authors may consider updating the percentage accordingly based on this more recent study.

Sibley, Margaret H., et al. "Variable patterns of remission from ADHD in the multimodal treatment study of ADHD." American Journal of Psychiatry 179.2 (2022): 142-151.

  1. Consider adding a brief conclusion paragraph after each subsection in the result section. 

Minor issues:

  1. Consider adding a column of neuroimaging methods in Table 2
  2. Typos and formatting issues:

a.     Line 74: add a comma between pharmacotherapy and preliminary

b.     Delete the additional row in Table 4

c.      Page 20, delete the empty rows. In addition, the line numbers are missing starting from Page 8.

  1. Ambiguity:

a.     Line 72: please specify “the number needed to treat”

b.     Line 96: change “very young subjects” to “young adult subjects”

Author Response

This well-written review by Capuzzi et al summarizes findings in biological predictors of adult ADHD treatment responses. Four broad types of biological predictors were surveyed and result summaries of these studies are nicely illustrated. Biomarkers of psychiatric disorders are at a rather early stage, and this review provides references for further investigation in adult ADHD treatment responses.

First of all, we thank the Reviewer for these nice comments and the interest in our manuscript.

Major issues:

  1. Line 43: “albeit up to 65% of affected children may continue to manifest symptoms in adulthood”.

According to Sibley et al, only 9.1% of the subjects demonstrated recovery by the study endpoint of 25 years of age. The authors may consider updating the percentage accordingly based on this more recent study. Sibley, Margaret H., et al. "Variable patterns of remission from ADHD in the multimodal treatment study of ADHD." American Journal of Psychiatry 179.2 (2022): 142-151.

We thank the Reviewer for this valuable comment. Accordingly, we added a comment in Introduction Section regarding the recent findings by Sibley et al. (2022)

  1. Consider adding a brief conclusion paragraph after each subsection in the result section.

Following the reviewer’s recommendation, we added a brief conclusion after each subsection of Result Section

Minor issues:

  1. Consider adding a column of neuroimaging methods in Table 2

Following the reviewer’s suggestion, we added a column detailing different brain imaging techniques

  1. Typos and formatting issues:
  1. Line 74: add a comma between pharmacotherapy and preliminary
  2. Delete the additional row in Table 4
  3. Page 20, delete the empty rows. In addition, the line numbers are missing starting from Page 8.
  1. Ambiguity:
  1. Line 72: please specify “the number needed to treat”
  2. Line 96: change “very young subjects” to “young adult subjects”

Following the Reviewer’s recommendations, we revised the manuscript.

Reviewer 2 Report

Overall, a very good review of the topic for adults with ADHD, with clinical relevance. The tables are helpful, and the written explanations are good.   Concerns:

  1. English writing - there are several spots - particularly in the introduction and discussion where the English grammar/writing is problematic. For example, the word 'till' should be 'until', and some sentences need editing to improve the English writing.
  2. On page 17, when discussing Duval et al, it is reported that "a low percentage of direction errors after the first MPH dose predicted remission after 6 months of pharmacotherapy." I am curious about the use of remission here. There is a lot written in the field of ADHD on remission, and the definition is not clear. Can the authors please clarify? How did this paper define remission (ie in this study, a 50% reduction on the ASRS was defined as remission, or whatever it is).
  3. On page 20, when discussing the Strauss et al paper, out of 28 drug free adult patients with ADHD, it is noted that only 8 were responders (and 20 were non responders). This is a very low response rate for MPH (8/28 = 28.6%). Please verify that this was reported properly, and was there any explanation as to why the response rate was so low compared to almost all other studies of MPH? Does this discredit the findings of this paper?
  4. In the final sentences of the paper, the authors discuss the 'three subtypes of the disorder'. In the DSM-IV, these were called subtypes. In the DSM-5, these are called 'presentations'. I suggest this be edited. 

Author Response

Overall, a very good review of the topic for adults with ADHD, with clinical relevance. The tables are helpful, and the written explanations are good.

First of all, we would like to thank the Reviewer for the interest in the manuscript.

Concerns:

  1. English writing - there are several spots - particularly in the introduction and discussion where the English grammar/writing is problematic. For example, the word 'till' should be 'until', and some sentences need editing to improve the English writing.

Following the reviewer’s recommendations, we improved the English writing.

  1. On page 17, when discussing Duval et al, it is reported that "a low percentage of direction errors after the first MPH dose predicted remission after 6 months of pharmacotherapy." I am curious about the use of remission here. There is a lot written in the field of ADHD on remission, and the definition is not clear. Can the authors please clarify? How did this paper define remission (ie in this study, a 50% reduction on the ASRS was defined as remission, or whatever it is).

We thank the Reviewer for this valuable comment. Accordingly, we added a comment regarding the definition of remission in the study by Duval et al. Moreover, we added a further comment at the end of Discussion section about the lack of universal definitions for response and remission in adult ADHD.

  1. On page 20, when discussing the Strauss et al paper, out of 28 drug free adult patients with ADHD, it is noted that only 8 were responders (and 20 were non responders). This is a very low response rate for MPH (8/28 = 28.6%). Please verify that this was reported properly, and was there any explanation as to why the response rate was so low compared to almost all other studies of MPH? Does this discredit the findings of this paper?

According to Reviewer’s comment, we have revised the paper byf Strauss et al. Strauss reported  that “the final study sample consisted of 28 initially non-medicated adult patients with ADHD, eight (28.6%) responders and 20 (71.4%) non-responders after a 4-week treatment with MPH”. The authors themselves reported that there was an unexpectedly low number of responders in the sample. However, they reported different explanations including the definition of response (a 30% decrease in the CAARS), the small sample size, the fact that patients suffered from significant psychiatric comorbidity and the short duration of the study (4 weeks).

  1. In the final sentences of the paper, the authors discuss the 'three subtypes of the disorder'. In the DSM-IV, these were called subtypes. In the DSM-5, these are called 'presentations'. I suggest this be edited. 

We certainly agree with this comment, and we have included the term “presentations” instead of “subtypes”.